# Positive-Unlabeled Learning for Control Group Construction in Observational Causal Inference

**Ilias Tsoumas**[1,2]  **Dimitrios Bormpoudakis**[1]  **Vasileios Sitokonstantinou**[2]
**Athanasios Askitopoulos**[1]  **Andreas Kalogeras**[1]  **Charalampos (Haris) Kontoes**[1]
**Ioannis Athanasiadis**[2]

[1]BEYOND Centre, IAASARS, National Observatory of Athens
[2]Artificial Intelligence Group, Wageningen University and Research
`{i.tsoumas, d.bormpoudakis, a.askitopoulos, a.kalogeras, kontoes}@noa.gr`
`{ilias.tsoumas, vasileios.sitokonstantinou, ioannis.athanasiadis}@wur.nl`

## Abstract

In causal inference, access to both treated and control units is essential for estimating treatment effects. When treatment assignment is random, the average treatment effect (ATE) can be estimated by comparing outcomes between groups. In non-randomized settings, techniques adjust for confounding to approximate the counterfactual and recover an unbiased ATE. A common challenge in observational studies is the absence of clearly labeled control units. To address this, we propose positive-unlabeled (PU) learning to identify control units from unlabeled data using only treated (positive) units. We evaluate this approach with simulated and real-world data, generating synthetic data from a causal graph to test the recovery of control groups for accurate ATE estimation. Applied to sustainable agriculture data, PU learning effectively distinguishes control units, enabling ATE estimates that closely match true effects. These results demonstrate PU learning's potential to enhance causal inference in settings lacking explicit control data. This work has important implications for observational causal inference, especially in fields like Earth, environmental, and agricultural sciences, where randomized experiments are costly and control units may be unavailable.

## 1 Introduction

In modern science, when a randomized controlled trial is not a feasible option, we turn to observational studies to answer causal questions. The most common type of these questions is of the form *"What is the effect of intervention $T$ on outcome $Y$?"*. The average treatment effect (ATE) is a popular quantity that we estimate to answer these types of questions. Inferring ATE estimates of an intervention on any outcome formally requires access to control and treated units. Equally important for observational causal inference is the capability to adjust for other variables that are involved as confounders in the intervention-outcome system of interest, so that any difference in outcome between control and treated groups can be attributed only to the intervention.

The absence of treated or control group data is a critical concern. Molinari [11] show how we can estimate the ATE if we do not know the treatment status for some of the units. Kuzmanovic et al. propose how to estimate conditional average treatment effects (CATE) [8] with missing treatment information. Beyond effect estimation under settings with partial absence of treatment information, Lancaster & Imbens [9], building on [18], show that even when some control units are mistakenly classified and may have received the treatment, what they refer to as "contaminated controls", it is still possible to recover unbiased estimates, as long as this misclassification is properly modeled in the analysis. Additionally, Rosenbaum & Rubin exploit the availability of a pool of potential controls

39th Conference on Neural Information Processing Systems (NeurIPS 2025) Workshop: Reliable ML from Unreliable Data.

on building a balanced control group using multivariate matching methods that incorporate the propensity score, ensuring similarity between treated and control units based on observed covariates [14].

In this work, we focus on the case of the total absence of units that are clearly labeled as controls, that is, units known not to have received the treatment. To address this issue, we propose the use of positive-unlabeled (PU) learning [1] as a formal framework for causal effect estimation in the absence of control units. Specifically, we identify, with high confidence, control units from a pool of unlabeled ones, using only the available treated (positive) units. Recently, PU learning has been proposed as a method to identify reliable control units from a pool of unlabeled ones [21]. In parallel, Kato et al. [6] propose a novel end-to-end PU learning method to estimate ATE in settings with a lack of control units. With this work, we contribute to this conceptual proposition through the following: i. We apply PU learning to infer a reliable control group from unlabeled data, independently of the downstream effect estimation process. This approach allows us to use any causal model and adjustment set for the effect estimation step, rather than relying on the end-to-end solution proposed by [6]. ii. We evaluate the performance of the PU learner when trained using a valid adjustment set defined by the back-door criterion, versus a more extensive set of features that includes known bad controls (e.g., mediators, colliders, and the outcome variable). Our results show that, despite violating causal assumptions, the latter configuration yields superior predictive performance. iii. We perform experiments with simulated and real-world data for applications in sustainable agriculture to showcase the usefulness of our approach. These contributions aim to showcase the PU learning as a useful addition to the observational causal inference toolbox, specifically when real-world interventions have taken place, but we lack labeled control units to realize an experiment. We emphasize its utility in the domains of Earth, environmental and agricultural sciences, where, given the availability of Earth observation and meteoclimatic data, it can unlock a plethora of quasi-experiments. We showcase this in our real-world examples.

## 2    Preliminaries & Problem Formulation

### 2.1    Observational Causal Inference

For each of the experimental setups that follow, simulated or real-world, we employ the relevant causal directed acyclic graph $G = (V, A)$ which includes all involved variables as vertices $V$ connected through directed edges $E$ and represent the causal relationships within the relevant treatment $T$ - outcome $Y$ system, e.g., the fertilizer - yield system. We limited all setups to binary treatments $T \in \{1, 0\}$ and we aim to estimate their unbiased effects on an outcome of interest $Y$. We choose the exact adjustment set $Z \subseteq V$, if any, that satisfies the back-door criterion relative to (T, Y), blocking every path from $T$ to $Y$ that contains an arrow into $T$, and no descendant nodes from $Z$ to $T$ are allowed. Then, we retrieve $ATE$ as shown in Eq. 1 based only on observational quantities.

$$
\begin{aligned}
\text{ATE} &= \mathbb{E}[Y \mid do(T = 1)] - \mathbb{E}[Y \mid do(T = 0)] \\
&= \sum_z \big( \mathbb{E}[Y \mid T = 1, Z = z] \\
&\quad - \mathbb{E}[Y \mid T = 0, Z = z] \big) \cdot P(Z = z)
\end{aligned}
\tag{1}
$$

Whether we express it using Equation 1 through a structural causal model [12], or via the potential outcomes framework [15], the formulation relies on certain fundamental assumptions that are shared, either explicitly or implicitly, by both frameworks:

$$
\textit{Unconfoundedness: } Y(t) \perp T \mid Z \quad \forall t \in \{0, 1\}
\tag{2}
$$

$$
\textit{Positivity: } \quad 0 < P(T = t \mid Z = z) < 1
$$
$$
\forall t \in \{0, 1\}, \forall z \in Z
\tag{3}
$$

$$
\textit{Consistency: } Y = Y(t) \quad \text{if } T = t
\tag{4}
$$

Firstly, the stable unit treatment value assumption (SUTVA) requires that there is (i) no interference between units, i.e., one unit's outcome is unaffected by other units' treatment, and (ii) well-defined treatment with no different forms of each treatment level. The unconfoundedness assumption (Eq. 2) requires that all confounders are measured, and conditioning on them removes bias. The positivity

assumption (Eq. 3) requires that every unit has a positive probability of receiving each treatment level given adjustment set $Z$. The consistency assumption (Eq. 4) requires that when we observe $T = t$, the observed outcome $Y$ is equal to the outcome that would result from an intervention setting $T$ to $t$ through the $do(\cdot)$ operator.

## 2.2 PU learning for control group construction

In this causal inference context, we first define the notation appropriately and adapt the standard assumptions of PU learning to the specific task of control group identification. Typically, PU learning aims to train a binary classifier using a set of labeled positive instances and a set of unlabeled instances that contain both positive and negative examples. In our work, we repurpose the PU learning framework to recover, from the unlabeled population and using only positively treated units, a subset of reliable negative (i.e., control) units for use in downstream effect estimation.

The target variable in our setting is the true class label, which corresponds to the treatment assignment $T \in \{1, 0\}$. Here, $T = 1$ indicates a truly treated (positive) unit and $T = 0$ indicates a truly control (negative) unit. Each unit is described by a feature vector $X$, which is used to estimate $T$. A key innovation we propose in this work is that $X$ is not restricted to the adjustment set $Z$; instead, our formulation allows $X$ to include any covariates in $V$, the set of observed variables in the causal graph $G$, that may help robustly estimate $T$, and even variables that are exogenous to the system described by $G$, if they are predictive. Thus, we allow the inclusion relationship $Z \subseteq V \subseteq X$. Unlike typical supervised learning, our setting introduces a third variable $S \in \{1, 0\}$, which is a label indicator. While $T$ represents the true (but possibly unobserved) treatment status, $S$ indicates whether a unit is labeled as treated. Specifically, $S = 1$ means the unit is observed/annotated as treated (positive), and $S = 0$ means the unit is unlabelled, and could be either a treated or a control unit.

$$\textit{SCAR:} \quad P(S = 1 \mid T = 1, X) = P(S = 1 \mid T = 1) = c$$
$$c \in (0, 1) \tag{5}$$

$$\textit{No Label Noise in Treated:} \ P(T = 0 \mid S = 1) = 0 \tag{6}$$

$$\textit{Controls in Unlabeled Set:} \ P(T = 0 \mid S = 0) > 0 \tag{7}$$

$$\textit{Separability:} \quad P(t = 1 \mid x) = \sigma(f(x)) \gg 0.5$$
$$\text{for most } x \in \text{Treated}$$
$$P(t = 0 \mid x) = \sigma(f(x)) \ll 0.5$$
$$\text{for most } x \in \text{Controls} \tag{8}$$

$$\textit{Smoothness:}$$
$$\|x - x'\| < \varepsilon, \text{ for small } \epsilon > 0 \tag{9}$$
$$\Rightarrow P(y = 1 \mid x) \approx P(y = 1 \mid x')$$

The selected completely at random (SCAR) assumption (Eq. 5) states that labeled positives are selected uniformly at random from all treated units, and selection for labeling is independent of features. The Eq. 6 implies that there is no possibility of mislabeling, e.g., a unit labeled as treated to be truly control. The Eq. 7 requires at least a control unit to be included in the unlabeled dataset. The separability assumption (Eq. 8) states that there exists a decision function $f(x)$ that reliably separates the two classes among unlabeled units via score thresholding (ie $\sigma(\cdot)$ denotes the sigmoid function). The smoothness assumption (Eq. 9) that two units with nearby feature vectors $x$ $x'$ shared the same treatment status, which forces us to allow post-treatment and exogenous features to be included in $X$. Finally, the ultimate goal of the PU learner in our framework is to estimate the treatment assignment probability $P(T = 1 \mid X)$. This enables the identification of units that, based on their features, are highly likely to be truly untreated (as formalized in Eq. 10), and can thus be used as reliable controls in causal effect estimation.

$$P(T = 0 \mid X) = 1 - P(T = 1 \mid X) \gg 0.5 \tag{10}$$

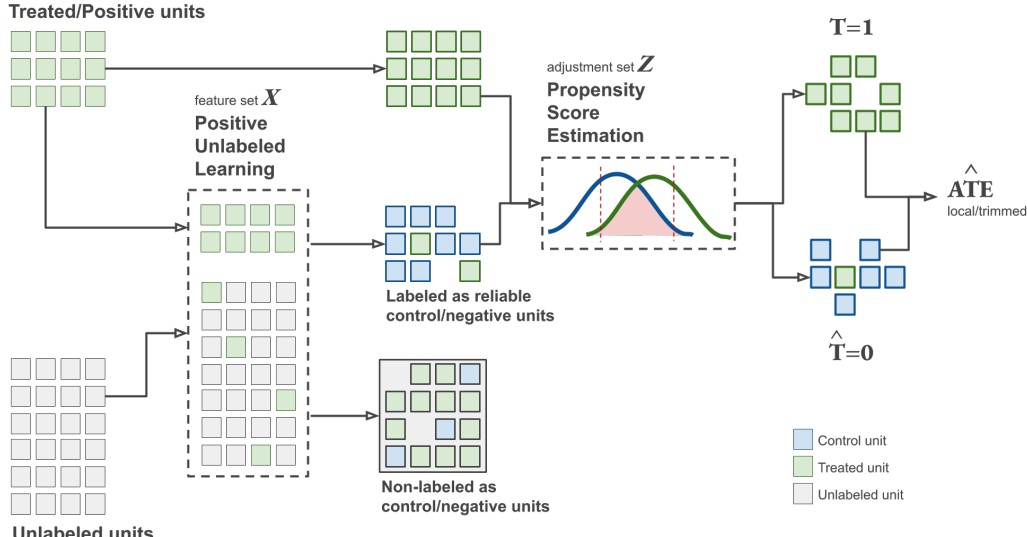

Figure 1: Overview of the use of PU learning as a preparatory step for causal estimation tasks that lack control groups.

# 3   Methodology & Experimentation

In this section, we provide a detailed explanation of: (i) how we use PU learning as a preparatory step to construct a reliable control group in cases where such a group is not readily available, thereby enabling causal effect estimation; and (ii) the four experimental setups, two based on simulated data and two on real-world datasets, used to evaluate our approach.

## 3.1   Framework for control group construction

As Fig. 1 illustrates, we emphasize cases where treated and unlabeled units are available, in single-training-set scenarios either with simulated or real-world datasets, because we focus mainly in Earth, environmental and agriculture cases where we are dealing with units of land (e.g., agricultural parcels) that belong on the same population/dataset and either they receive an intervention/event or we do not know if they receive it or not. Specifically, we use a two-step technique. In the first step, we leverage the SPY method [10] where some of the real treated units are turned into spies, which we "hide" in the unlabeled dataset. A Naive Bayes classifier is trained, considering the unlabeled units as controls. Then, we label as "reliable" control units those for which the posterior probability is lower than the lowest of spies. In the second step, we employ iterative SVM (iSVM) [22]. In each iteration, an SVM classifier is trained using the real treated units and the reliable controls from the first step. The unlabeled units that are classified as controls by this classifier are then added to the set of reliable controls for the next iteration, till no class change occurs between two iterations. Thus, we categorize the unlabeled units into two groups: the reliable control units, in which the PU learning assigned the control label with a tunable confidence, and the group with units that were left unlabeled because they do not differ enough from the truly treated, given the features $X$. We do not consider the latter as treated because we already have a group with confirmed truly treated units, so there is no reason to risk adding bias with some possible falsely labeled units as treated. Naturally, these two new groups will probably contain some falsely labeled units.

Next, we train a propensity score estimator (e.g., logistic regression) using the truly treated units and the reliable controls identified by the PU learner. The adjustment set $Z$, which satisfies the backdoor criterion based on the causal graph $G$ representing the system, is preselected for estimating the effect of $T$ on $Y$. Consistent with [4], we train the propensity score model using only the covariates in $Z$, which includes pre-treatment variables and proxies for the selection mechanism, while excluding the outcome variable and post-treatment variables such as mediators and colliders. We plot the propensity scores per group and we trim the scores of the two groups to ensure sufficient overlap between them. The trimming ensures that we avoid an extrapolation in regions with little comparability between

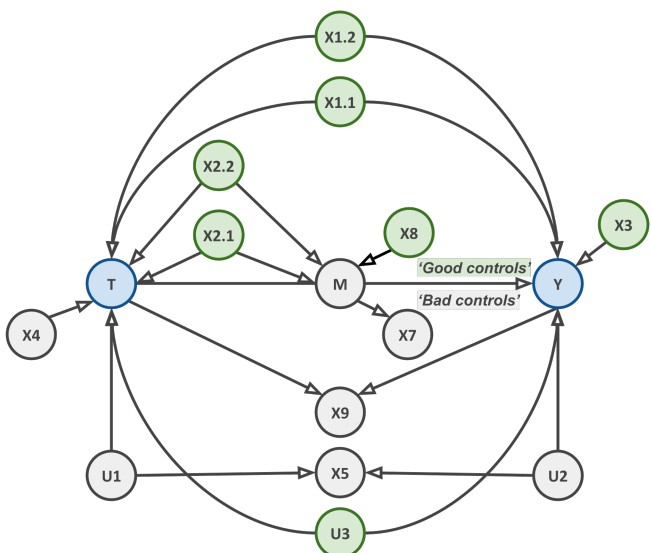

Figure 2: The causal graph $G_{sim}$ serves as the data-generating process for both linear and non-linear simulations.

| Dataset | PU method | Feature Set | # Positives (true treated) | # Spies | # Unlabeled | | Evaluation Metrics | | | | | | | |
| | | | | | controls | treated | # Selected | | # Non-selected | | Control Recall | Control Precision | Contamination Rate | Treated Leakage |
| | | | | | | | # Controls | # Treated | # Controls | # Treated | | | | |
| Linear | SPY | Z | 346 | 103 (30%) | 505 | 149 | 334 | 31 | 171 | 118 | 0.661 | 0.915 | 0.085 | 0.208 |
| | | X | | | | | 505 | 28 | 0 | 121 | 1.000 | 0.947 | 0.053 | 0.188 |
| | SPY+iSVM | Z | | | | | 143 | 4 | 362 | 145 | 0.283 | 0.973 | 0.027 | 0.027 |
| | | X | | | | | 505 | 0 | 0 | 149 | 1.000 | 1.000 | 0 | 0 |
| Non-linear | SPY | Z | 365 | 109 (30%) | 478 | 157 | 186 | 28 | 292 | 129 | 0.389 | 0.870 | 0.131 | 0.178 |
| | | X | | | | | 476 | 15 | 2 | 142 | 0.996 | 0.969 | 0.031 | 0.095 |
| | SPY+iSVM | Z | | | | | 121 | 16 | 357 | 141 | 0.253 | 0.883 | 0.117 | 0.102 |
| | | X | | | | | 474 | 0 | 4 | 157 | 0.992 | 1.000 | 0 | 0 |
| Sowing | SPY | Z | 35 | 5 (15%) | 121 | 15 | 92 | 3 | 29 | 12 | 0.760 | 0.968 | 0.032 | 0.200 |
| | | X | | | | | 72 | 4 | 49 | 11 | 0.595 | 0.947 | 0.053 | 0.267 |
| | SPY+iSVM | Z | | | | | 97 | 5 | 24 | 10 | 0.802 | 0.951 | 0.049 | 0.333 |
| | | X | | | | | 101 | 5 | 20 | 10 | 0.835 | 0.953 | 0.047 | 0.333 |
| Fertilization | SPY | Z | 23 | 3 (15%) | 99 | 11 | 64 | 3 | 35 | 8 | 0.646 | 0.955 | 0.045 | 0.273 |
| | | X | | | | | 71 | 5 | 28 | 6 | 0.717 | 0.934 | 0.066 | 0.455 |
| | SPY+iSVM | Z | | | | | 89 | 7 | 10 | 4 | 0.899 | 0.927 | 0.073 | 0.636 |
| | | X | | | | | 87 | 8 | 12 | 3 | 0.878 | 0.916 | 0.084 | 0.727 |

Table 1: Summary of control group selection using SPY and SPY+iSVM across four datasets (Linear, Non-linear, Sowing, Fertilization), evaluated with feature sets $Z$ (adjustment set) and $X$ (the most informative variables set about treatment).

groups, and we focus on $ATE$ estimation on the region of common support. Given that we have recovered as reliable controls the units that, intuitively speaking, significantly differ from the real treated for the PU learning estimator, which is trained on $X$ and in parallel we estimate propensity scores training an estimator on $Z$, it is clear that we risk drastic shrinkage of units in the area of overlap between the two groups due to the significant overlap of feature sets $Z \subseteq X$. This area of overlap is prone to be the area of the smallest propensity scores, given that reliable controls are units that are based on $X$ place in a conceptually similar area from the PU learning estimator. Consequently, an issue arises about how strong the positivity assumption holds each time and what exactly this strong *'trimmed'* or *'local'* $ATE$ quantity represents. Finally, we estimate the $ATE$ with several estimators using the trimmed, based on propensity scores, treated and reliable control groups for reasons of completeness and framework assessment. We use linear regression and distance matching as baseline estimation methods, and Inverse Propensity Score Weigthing (IPW) [19] and the causal machine learning method T-learner [7].

## 3.2 Experimental setups & data

We simulate data under two different assumptions (generating $n = 1000$ samples for each), one with linear and one with non-linear causal relationships, to test our ideas in a fully controllable environment. For this purpose, we construct the causal graph $G_{sim}$ of Fig. 2, where we employ the Cinelli et al. work [2] to introduce $G_{sim}$ a wide range of different relationships *'confounders'*, *'mediators'*,

'colliders'. Specifically, vertices $V_{good} = \{X_{1.1}, X_{1.2}, X_{2.1}, X_{2.2}, U_3\}$ are 'good controls' reducing bias by blocking back-door paths if we control for them. The $V_{bad} = \{X_3, X_5, X_7, X_9, M\}$ are 'bad controls', with addition of $X_3$ as control would lead to 'bias amplification'. The $X_5$ are known as 'M-bias' that induce bias through $U_1, U_2$. $M$ mediates the effect we want to estimate, so we keep this path untouched without controlling for this variable, similarly to $X_7$, where a control for it is equivalent to a partial control for mediator $M$. $X_9$ is a typical 'bad control' because controlling for it opens a colliding path and induces 'selection bias'. Finally, vertices $V_{neutral} = \{X_3, X_8\}$ are neutral in terms of inserting or removing bias but can be useful controls in terms of $ATE$ precision. Thus, as expected from the structure of the causal graph, the minimum adjustment set $Z_{sim}$ that satisfies the back-door criterion contains the 'good controls' so $Z_{sim} = V_{good} = \{X_{1.1}, X_{1.2}, X_{2.1}, X_{2.2}, U_3\}$ for an unbiased estimation of $ATE$ of treatment $T$ on outcome $Y$ of the causal system $G_{sim}$. Section B of the Appendix presents the data-generating process in detail.

Also, we investigate the applicability of our ideas in two real-world applications in sustainable agriculture. First, we use the causal graph, the data and the causal effect estimation setup from Tsoumas et al. [20]. They estimate the $ATE$ of treatment $T$: whether the cotton field is sown on a recommended day by a smart farming system, on the outcome $Y$, the final yield of the field. We use the adjustment set $Z_{sowing} = \{Weather\ on\ sowing\ date,\ Soil\ moisture\ on\ sowing\ date,\ Topsoil\ properties,\ Topsoil\ organic\ carbon,\ Seed\ variety,\ Geometry\ of\ field\}$ that satisfies the back-door criterion of their proposed causal graph, and also we use the additional available variables $V_{extra} = \{Crop\ growth,\ Location\ \&\ area\ of\ field,\ Harvest\ date\ \&\ cultivation\ period,\ Yield\}$ that are not used for effect estimation to test the use of an expanded feature set $X_{sowing} = Z_{sowing} + V_{extra}$ over the adjustment set $Z_{sowing}$ to construct a group of reliable control units and the hold of the positivity assumption. As a second real-world scenario, we introduce a new dataset about the impact of a digestate fertilizer (i.e., a sustainable alternative to chemical fertilizers) on wheat biomass in fields cultivated with durum wheat. In this work, we adopt the well-known fertilization–yield causal graph example from Pearl's book [13] and enrich it using the more detailed ground-truth causal graph on soil processes from [17]. In this dataset, the adjustment and feature sets are defined as follows: $Z_{fertilizer} = \{Topsoil\ organic\ carbon,\ Accumulated\ precipitation,\ Growing\ degree\ days,\ Accumulated\ vegetation\ moisture\ content,\ Soil\ moisture,\ Soil\ type\ Seed\ variety\}$ and $X_{fertilizer} = Z_{fertilizer} + \{Exogenous\ organic\ matter\ indexes,\ Produced\ biomass\ proxy\}$. More details about the effect estimation and the data in Table 1 in Sec. A of the Appendix.

## 4 Results

In this section, we present and discuss the results of our experiments on the proposed use of PU learning as a means to enable causal effect estimation in settings where a confirmed control group is entirely absent, but a source of unlabeled units is available.

### 4.1 Evaluation through Classification metrics

To evaluate the framework, we emulate PU conditions using the popular 'hide and seek' approach [16]. Specifically, we construct positive–unlabeled datasets by transforming the original simulated and real-world positive–negative datasets. We do this simply by hiding a percentage of positive/treated units within the negative/control units, something that allows us to consider this mixed group with units from both treatment classes as an unlabeled set to test our ideas. This satisfies the SCAR assumption (Eq. 5) and facilitates the assessment of the PU learner using common binary classification evaluation metrics. However, we have to clarify some conceptual changes from typical classification metrics that better align with and reflect a causal inference context. While throughout the paper we refer to treated units as positives and controls as negatives in the PU learning setup, during evaluation, we treat true controls as the positive class to assess how well the method recovers clean/reliable control units. Thus, for clarity, under this evaluation perspective, True Positives (TP) correspond to true control units correctly recovered as controls, False Positives (FP) correspond to treated units wrongly selected as controls, False Negatives (FN) to true control units that were not recovered and True Negatives (TN) to treated units correctly not selected (held out as no control units). Thus, we use: the Control Recall equation (Eq. 11) to assess how many of the true controls were recovered (it measures the coverage of the control group); the Control Precision equation (Eq. 12), which summarizes how many were actual controls out of selected as 'reliable controls'; the Contamination Rate (Eq. 13), which refers to the proportion of selected as 'reliable controls' that were treated; and we introduce

the Treated Leakage metrics (Eq. 14) that measures how many units mistakenly ended up in 'reliable controls' among true treated units.

$$\text{Control Recall} \ = \frac{TP}{TP + FN} \tag{11}$$

$$\text{Control Precision} \ = \frac{TP}{TP + FP} \tag{12}$$

$$\text{Contamination Rate} \ = \frac{FP}{TP + FP} \tag{13}$$

$$\text{Treated Leakage} \ = \frac{FP}{FP + TN} \tag{14}$$

In Table 1, the evaluation metrics of PU learning for retrieving real control units from an unlabeled pool of mixed control and treated units are presented in detail for the four different datasets. In the two simulated, linear and non-linear, datasets, our results show that the combination of SPY with iSVM returns the best results in all metrics with the use of the $X$ feature set instead of the $Z$ adjustment set. The use of $X$ extremely outperforms in any metric, with more significant gains in the almost elimination of any leakage of treated in the 'reliable controls' and the increase of recall. Even the SPY method alone using $X$ outperforms the combined solution of SPY and iSVM when they are trained on $Z$. In the sowing and fertilization datasets, recall and precision follow almost the same fluctuations as in the simulated data. However, leakage of treated units presents a slight increase with the use of the feature set $X$ in comparison with the adjustment set $Z$. Furthermore, the fertilization dataset, when SPY and iSVM are trained on $X$, presents very high treated leakage, with 8 out of 11 hidden treated units classified as controls, which risks significantly biasing the results of the following effect estimation task. A likely explanation is the very small sample size of spies and hidden treated units for the machine learning task.

| Dataset | Control Units | (treated, control) | true effect | PU Feature Set | Causal Effect Estimation Methods | | | | | | | | | | | |
|---|---|---|---|---|---|---|---|---|---|---|---|---|---|---|---|---|
| | | | | | Linear Regression | | | IPS weighting | | | Matching | | | T-Learner(RF) | | |
| | | | | | ATE | CI | p-value | ATE | CI | p-value | ATE | CI | p-value | ATE | CI | p-value |
| Linear | real controls | (495, 505) - no trim | | - | 3.664 | (3.447, 3.895) | 0.000 | 3.848 | (3.451, 4.237) | 0.000 | 3.927 | (3.601, 4.192) | 0.001 | 4.029 | (3.939, 4.119) | - |
| | SPY | (36, 232) | 3.000 | Z | 2.718 | (2.250, 3.143) | 0.000 | 3.081 | (1.927, 4.121) | 0.001 | 3.261 | (2.648, 3.620) | 0.001 | 3.520 | (3.211, 3.828) | - |
| | | (450, 414) | | X | 2.800 | (2.631, 2.997) | 0.000 | 2.818 | (2.337, 3.376) | 0.001 | 3.282 | (2.989, 3.392) | 0.001 | 3.529 | (3.428, 3.630) | - |
| | SPY+iSVM | (22, 147) | | Z | 2.757 | (2.109, 3.338) | 0.000 | 3.231 | (1.817, 4.500) | 0.001 | 3.667 | (2.906, 4.250) | 0.001 | 3.771 | (3.331, 4.212) | - |
| | | (414, 424) | | X | 3.021 | (2.861, 3.185) | 0.000 | 3.188 | (2.765, 3.559) | 0.001 | 3.574 | (3.320, 3.726) | 0.001 | 3.814 | (3.720, 3.909) | - |
| Non-linear | real controls | (522, 478) - no trim | | - | 9.717 | (9.367, 10.094) | 0.000 | 9.799 | (9.296, 10.229) | 0.001 | 9.597 | (9.155, 9.869) | 0.001 | 9.829 | (9.646, 10.011) | - |
| | SPY | (17,77) | 9.525 | Z | 8.057 | (5.989, 10.452) | 0.000 | 7.775 | (4.819, 10.827) | 0.001 | 7.209 | (4.660, 8.793) | 0.001 | 7.261 | (6.322, 8.201) | - |
| | | (513, 487) | | X | 9.480 | (9.100, 9.886) | 0.000 | 9.570 | (9.040, 10.035) | 0.001 | 9.321 | (8.900, 9.607) | 0.001 | 9.324 | (9.132, 9.516) | - |
| | SPY+iSVM | (86, 137) | | Z | 7.962 | (6.762, 8.993) | 0.000 | 7.884 | (6.269, 9.466) | 0.001 | 7.556 | (6.318, 8.178) | 0.001 | 7.877 | (7.384, 8.371) | - |
| | | (513, 470) | | X | 9.719 | (9.378, 10.0953) | 0.000 | 9.820 | (9.369, 10.254) | 0.001 | 9.805 | (9.446, 10.092) | 0.001 | 9.887 | (9.701, 10.073) | - |
| Sowing | real controls | (50, 121) | | - | 546 | (211, 880) | 0.002 | 471 | (138, 816) | 0.001 | 448 | (186, 760) | 0.006 | 372 | (215, 528) | 0.024 |
| | SPY | (13, 92) | expected positive | Z | 211 | (-479, 902) | 0.544 | 315 | (-487, 1024) | 0.138 | 244 | (-733, 1099) | 0.226 | 259 | (-134, 653) | - |
| | | (14, 14) | | X | 664 | (-267, 1596) | 0.147 | 811 | (122, 1660) | 0.030 | 689 | (-54, 1369) | 0.063 | 777 | (376, 1178) | - |
| | SPY+iSVM | (25, 43) | | Z | 156 | (-463, 776) | 0.615 | 322 | (-25, 724) | 0.087 | 293 | (-145, 723) | 0.128 | 306 | (82, 530) | - |
| | | (39, 47) | | X | 466 | (-61, 993) | 0.082 | 524 | (170, 992) | 0.029 | 480 | (132, 857) | 0.015 | 390 | (195, 586) | - |
| Ferilization | real controls | (34, 99) - no trim | | - | 1.188 | (-0.386, 2.762) | 0.138 | 1.160 | (-0.094, 2.649) | 0.055 | 0.708 | (-0.834, 2.266) | 0.200 | 1.104 | (-0.484, 1.725) | - |
| | SPY | (8, 38) | expected positive | Z | 3.000 | (-0.600 6.599) | 0.097 | 1.866 | (-0.173, 4.351) | 0.094 | 1.111 | (-1.188, 2.842) | 0.219 | 0.903 | (-0.141, 1.947) | - |
| | | (20, 47) | | X | 2.342 | (-0.168, 4.851) | 0.067 | 0.805 | (-0.711, 2.604) | 0.167 | 0.425 | (-1.173, 2.268) | 0.345 | 0.464 | (-0.363, 1.292) | - |
| | SPY+iSVM | (24, 67) | | Z | 1.302 | (-0.566, 3.170) | 0.168 | 1.190 | (-0.391, 2.843) | 0.076 | 0.981 | (-0.426, 3.142) | 0.162 | 0.542 | (-0.228, 1.312) | - |
| | | (31, 67) | | X | 1.750 | (0.003, 3.498) | 0.049 | 1.322 | (-0.347, 3.122) | 0.047 | 0.742 | (-0.555, 2.265) | 0.192 | 0.823 | (0.158, 1.488) | - |

Table 2: Effect estimation using treated units and (i) real control, (ii) retrieved reliable controls under various PU learning methods, feature sets, and ATE estimation techniques.

In all datasets, the number of retrieved 'reliable controls' tends to be larger if the larger $X$ feature set is used for PU learning, see column *'# Selected'* in Table 1. After the trimming of both groups' units based on propensity scores to secure treated and control units have overlapped propensities, it appears (in the column *'(treated, control)'* of Table 2) that bigger overlap between groups emerges in the more complex 2-steps setup (SPY plus iSVM with the use of $X$ feature set). Fig. 3, based on the nonlinear dataset, illustrates precisely that the feature set $X$ enables reliable controls to be more evenly distributed across the full range of propensity scores, in contrast to propensity score estimators using the adjustment set $Z$, which result in a more skewed distribution of controls. This is also visualized in Sec C of the Appendix, where the propensity scores overlap for all datasets are depicted. Due to the skewed propensity scores distribution of inferred reliable controls even for the best case (i.e., SPY plus iSVM with $X$) of sowing and fertilization datasets, we apply asymmetric trimming to retain units with estimated propensity scores in the range $[0.1, 0.6]$. This decision ensures that we operate within a region of the covariate space where there is reasonable overlap between positively labeled treated units and inferred control candidates. However, this approach limits our estimand to an $ATE$ within this low- to low-to-moderate-propensity subpopulation. Consequently, our $ATE$ estimates are not directly generalizable to the full population or to high-propensity treated

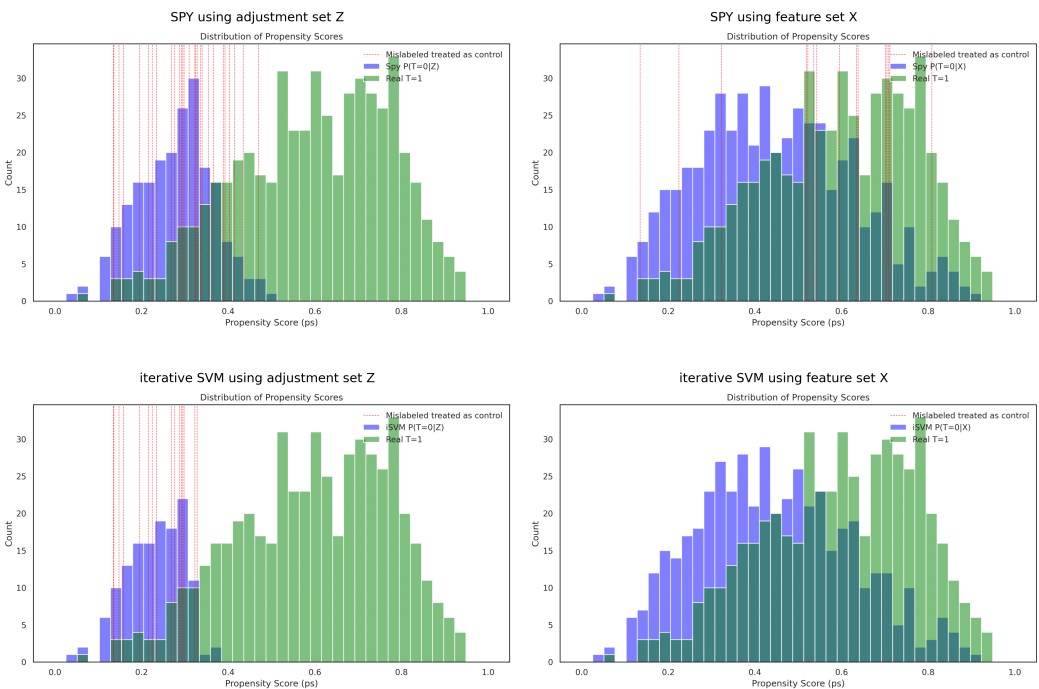

Figure 3: Propensity scores of 4 different combinations on non-linear experimental dataset.

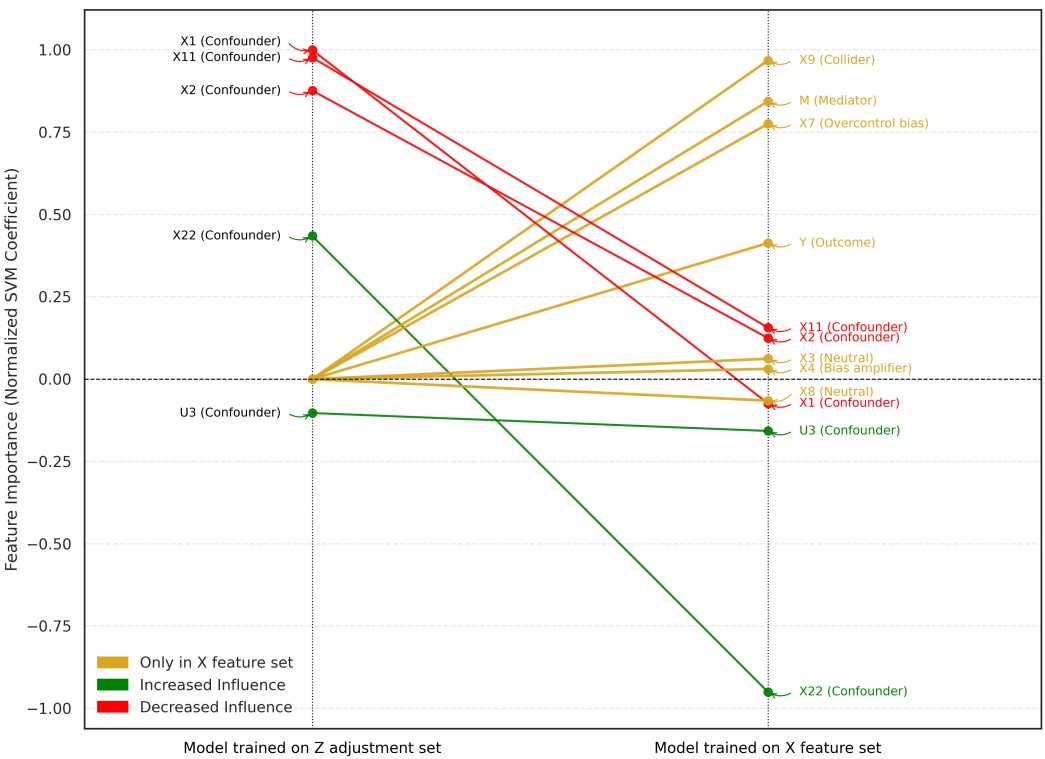

Figure 4: Slope chart illustrating changes in normalized SVM coefficients when training on the $Z$ adjustment set compared to the full $X$ set.

units, which are trimmed out due to a lack of comparable control analogs. While this trimming reduces the risks of extrapolation and positivity violation, it introduces a form of selection bias and may underestimate the true heterogeneity of treatment effects across the broader covariate space. So, in terms of proper classification/reliable controls identification, the full 2-step PU learning method outperforms the use of SPY alone in most cases, and the superiority of using of $X$ feature set instead of the $Z$ adjustment set for the PU learning is profound. The slope chart of Fig 4, regarding the nonlinear dataset, highlights how the inclusion of additional 'bad control' features from the $X$ feature set can critically affect the accuracy of the PU model. Specifically, post-treatment variables acting as colliders and mediators, which should be excluded from an effect estimation task [4], are instead ranked as highly informative features when included for reliable control group identification. In Sec. D of the Appendix, we included a slope chart per dataset that compares the top-15 features in terms of influence and their change between the iSVM trained on $Z$ and $X$, using model coefficients as feature importance.

## 4.2 Evaluation through ATE estimation

As a final evaluation step, since control units are available in reality for each dataset, we leverage various methods from different methodological backgrounds to estimate the $ATE$ of each treatment on the outcome of interest, both before and after retrieving control units using our proposed approach. Observing the results in Table 2, we find that, for both simulated datasets and the sowing dataset, the setup using SPY and iSVM models trained on the full feature set $X$ successfully retrieves a statistically significant ATE that closely matches the ATE estimated using the confirmed treated and control groups across all causal estimators. We also observe that when PU learning is performed using the adjustment set $Z$, the estimated ATE tends to be underestimated and lacks statistical significance. This is likely due to the aggressive trimming threshold $[0.1, 0.3]$, which restricts the analysis to a narrow subpopulation and results in a very small number of remaining units in both the treated and control groups.

For the fertilization dataset, the results are less stable (Table 2). Using the real control units, we observe a positive ATE, though not statistically significant. After applying the PU learning method, specifically the more robust SPY plus iSVM plus $X$ scenario, we observe a slight increase in ATE estimates, with linear regression and IPW yielding statistically significant results ($p$-value < 0.05), while matching and the T-Learner produce comparable, though less stable, estimates. However, these findings should be interpreted with caution due to the high level of treated unit leakage in this scenario. Additionally, we applied our method to a real, unlabeled dataset, enriched with available confirmed controls, to identify reliable units and estimate the ATE using only these controls (ignoring their actual treatment status) and the treated group. This dataset includes 616 unlabeled units and 99 confirmed controls. Using the SPY plus iSVM plus $X$ method and trimming the propensity scores to the range $[0.05, 0.6]$ to ensure overlap, we identified 67 reliable controls, 55 of which are confirmed controls. Once again, the results align with those obtained using only real controls or engineered pseudo-unlabeled data (Table 2). Specifically, for the real unlabeled dataset: linear regression estimated an $ATE$ of 1.641 ($CI : [-0.393, 3.675], p = 0.112$); IPW estimated an $ATE$ of 1.255 ($CI : [-0.613, 3.002], p = 0.113$); matching an $ATE$ of 1.207 ($CI : [-0.709, 2.528], p = 0.133$); and the T-Learner an $ATE$ of 0.877 ($CI : [0.075, 1.679]$). Overall, even in this less reliable setting of the fertilization dataset, the results suggest that our PU learning approach can effectively recover the true $ATE$ in the absence of a known control group.

## 5 Conclusions

In this work, we proposed, implemented, and evaluated the use of PU learning as a reliable approach for identifying control units from unlabeled data for causal effect estimation, addressing the challenge of not having known controls. Our experiments demonstrate that, under appropriate configurations, such as avoiding restrictions on effect estimation covariates when training the PU learner, this approach can be effective. Further work should examine the trimming–extrapolation trade-off, explore using $P(T = 1|X)$ to enrich the treated group, apply more advanced PU methods, and test on causal effect estimation benchmarks. Ultimately, we provide a simple, practical, and realistic solution that can unlock a wide range of quasi-experiments in Earth, environmental, and agricultural sciences, especially given the growing availability of large-scale Earth observation data.

## Acknowledgments and Disclosure of Funding

We express our gratitude to Alexis Apostolakis for his informative presentation on PU learning during one of our journal club gatherings, which served as the initial spark for the conception of this idea. We also thank Selected Biogas Farsala SA for their collaboration and provision of data for one of the use cases. This work was supported by the project "Climaca" (ID: 16196), carried out within the framework of the National Recovery and Resilience Plan Greece 2.0, funded by the European Union – NextGenerationEU.

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

## A  Real-World Datasets

For the sowing case, we follow the causal graph construction presented in Tsoumas et al. [20], which guides the selection of relevant variables. In the fertilization case, we model the effect of fertilizer application on biomass production during the final phenological stages, incorporating both decisional and biological confounders. These confounders are selected based on a merged causal graph that combines the canonical fertilization–yield graph from Pearl's book [13] with a more detailed soil-process causal graph from [17]. Additionally, exogenous organic matter indexes, included as $X$ variables, are retrieved from the literature as features indicative of fertilization activity detectable by Sentinel-2 [3, 5]. As detailed in Table 3, adjustment variables ($Z$) cover the cultivation period up to 01/05/2023, the treatment variable ($T$) indicates whether fertilizer was applied at least once between 01/09/2022 and 01/05/2023, and the outcome ($Y$) corresponds to NDVI-derived trapezoidal area between 01/05/2023 and 30/06/2023, capturing the produced biomass at the end of the season.

| Sowing | | | | Fertilization | | | |
|---|---|---|---|---|---|---|---|
| Feature(s) name | Variable/Vertex | Source | Set | Feature(s) name | Variable/Vertex | Source | Set |
| LOW, HIGH | Weather on sowing date | Nearest weather station | Z | SOC_prediction | Topsoil organic carbon (2022) | NOA ML model | Z |
| peak_ndvi, trapezoidal_ndvi_sow2harvest | Crop growth | NDVI via Sentinel-2 | X | accum_precip_total_m | Accumulated precipitation (2022-09-01 - 2023-05-01) | ERA5-land | Z |
| ndwi_sowingday | Soil moisture on sowing date | NDWI via Sentinel-2 | Z | gdd_base_0 | Growing degree days (2022-09-01 - 2023-05-01) | ERA5-land | Z |
| clay_mean, silt_mean, sand_mean | Topsoil properties | Map by ESDAC | Z | ndmi_trapezoidal_area | Accumulated vegetation moisture content (2022-12-01 - 2023-05-01) | NDMI via Sentinel-2 | Z |
| occont_mean | Topsoil organic carbon | Map by ESDAC | Z | sum_soil_moisture_mean | Soil moisture (2022-09-01 - 2023-05-01) | ERA5-land | Z |
| var_{ST_402, ..., ELPIDA} | Seed variety | Farmers' Cooperative | Z | st_{Cambisols, ..., st_Luvisols} | Soil type | SoilGrids World Reference Base(2006) Soil Groups | Z |
| ratio | Geometry of field | Farmers' Cooperative | Z | POI_{7807, ..., 10486} | Seed variety | LPIS from NPA | Z |
| lat, lon, perimeter, field_area | Location & area of field | Farmers' Cooperative | X | eomi_{1-4}, nbr2 - stats: max, min, mean, std, skew | Exogenous organic matter indexes | via Sentinel-2 | X |
| hday_sin, hday_cos, len_season | Harvest date & cultivation period | Farmers' Cooperative | X | | | | |
| prediction | Treatment (sown on recommended or not date) | Farmers' Cooperative, RS | T | TREATMENT | Treatment (apply or not fertilizer) | Fertilizer company | T |
| yield21 | Outcome (Yield) | Farmers' Cooperative | X | ndvi_trapezoidal_area | Outcome (produced biomass proxy for 2023-05-01 - 2023-06-30) | NDVI via Sentinel-2 | X |

Table 3: Variables used for causal modeling in the sowing and fertilization case studies, categorized by their role (Z: adjustment set, X: features set, T: treatment, Y: outcome) and source.

## B  Simulation Datasets

We simulate data under two different structural assumptions. A **linear** setup and a **nonlinear** setup. In both cases, we generate $n = 1000$ samples.

### Observed, Unobserved and Latent Variables

$$X_1, X_2, X_{11}, X_{22}, X_3, X_4, X_8 \sim \mathcal{N}(0,1)$$
$$U_1, U_2, U_3 \sim \mathcal{N}(0,1)$$
$$X_5 = 0.6U_1 + 0.6U_2 + \varepsilon_5, \quad \varepsilon_5 \sim \mathcal{N}(0, 0.1^2)$$

### Linear Setup

### Treatment Assignment

$$\text{logit}\left(P(T=1)\right) = 1.2X_1 + 0.8X_2 + 1.4X_{11} + 0.6X_{22} + 0.7X_4 + 1.0U_1 + 1.0U_3$$
$$T \sim \text{Bernoulli}\left(\sigma(\cdot)\right), \quad \sigma(z) = \frac{1}{1 + e^{-z}}$$

### Mediator

$$M = 1.5T + 0.7X_8 + 0.5X_2 + 0.3X_{22} + \varepsilon_M, \quad \varepsilon_M \sim \mathcal{N}(0, 0.1^2)$$

### Outcome

$$Y = 2.0M + 0.7X_1 + 0.5X_{11} + 0.6X_3 + 1.0U_2 + 1.0U_3 + \varepsilon_Y, \quad \varepsilon_Y \sim \mathcal{N}(0, 0.1^2)$$

### Proxy Variables

$$X_7 = 1.2M + \varepsilon_{X_7}, \quad \varepsilon_{X_7} \sim \mathcal{N}(0, 0.1^2)$$
$$X_9 = 1.0T + 1.0Y + \varepsilon_{X_9}, \quad \varepsilon_{X_9} \sim \mathcal{N}(0, 0.1^2)$$

**Nonlinear Setup**

**Treatment Assignment**

$$\text{logit}\left(P(T=1)\right) = 1.2\tanh(X_1) + 0.8\sin(X_2) + 1.4\tanh(X_{11}) + 0.6\sin(X_{22})$$
$$+ 0.7\tanh(X_4) + 1.0U_1 + 0.8U_3X_1$$
$$T \sim \text{Bernoulli}\left(\sigma(\cdot)\right)$$

**Mediator**

$$M = 1.5T + 0.7\sqrt{|X_8|} + 0.5\log(1+|X_2|) + 0.3X_{22} + 0.1X_2X_8 + \varepsilon_M$$

**Outcome**

$$Y = 2.0M^2 + 0.7X_1 + 0.5X_{11} + 0.6\sin(X_3) + 0.2X_3^2 + 1.0U_2 + 1.0U_3 + \varepsilon_Y$$

**Proxy Variables**

$$X_7 = 1.2M + \varepsilon_{X_7}$$
$$X_9 = 1.0T + 1.0Y + \varepsilon_{X_9}$$

In both setups, the variables $X_7$ and $X_9$ serve as proxies. $X_7$ for the mediator $M$, and $X_9$ as a collider involving both $T$ and $Y$.

## C  Propensity scores

Following the figures for each experimental dataset with an overview of how propensity score overlap varies in the current experimental dataset when using different predictor sets (i.e $X, Z$) and methods (i.e. SPY, SPY + iSVM). This highlights how the choice of predictor variables and estimation approach affects the positivity assumption and common support.

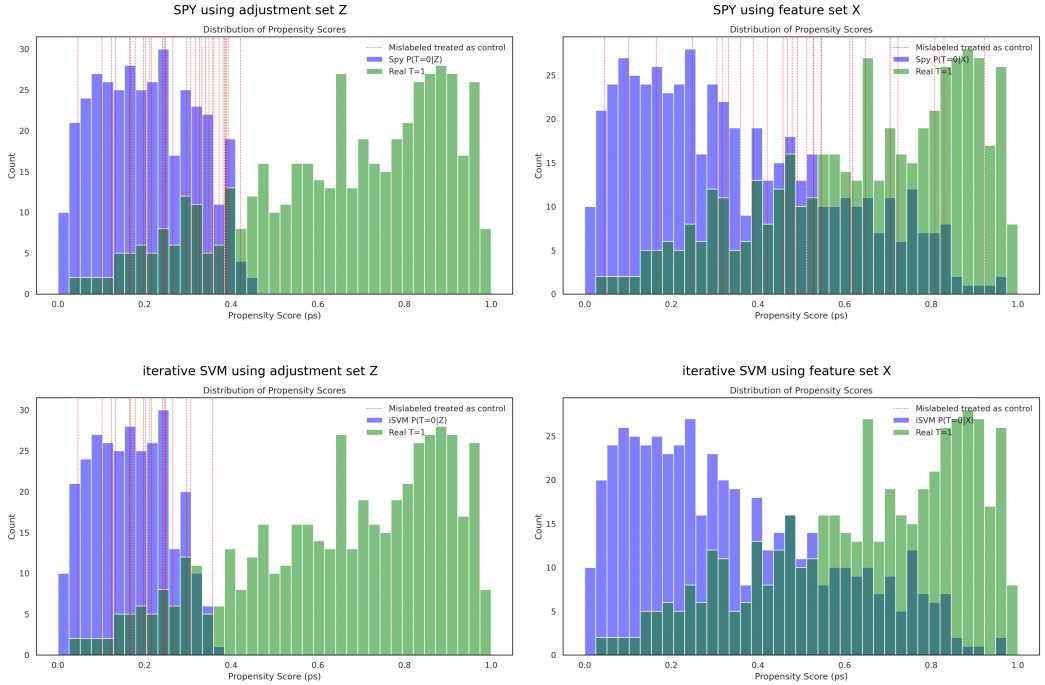

Figure 5: Propensity scores of 4 different combinations on linear experimental dataset

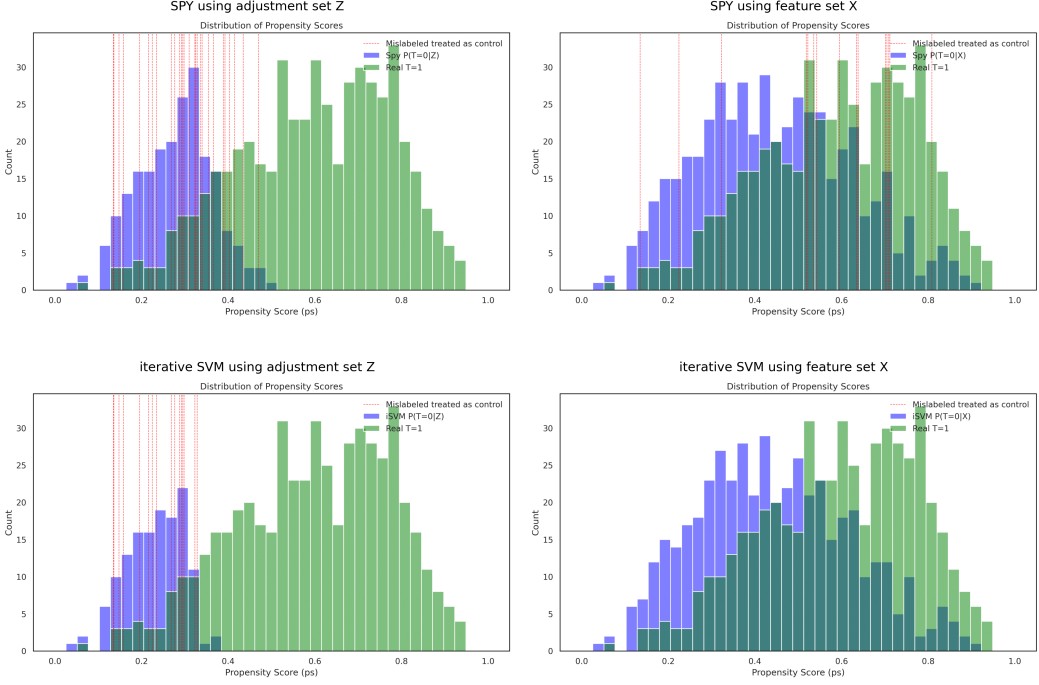

Figure 6: Propensity scores of 4 different combinations on non-linear experimental dataset

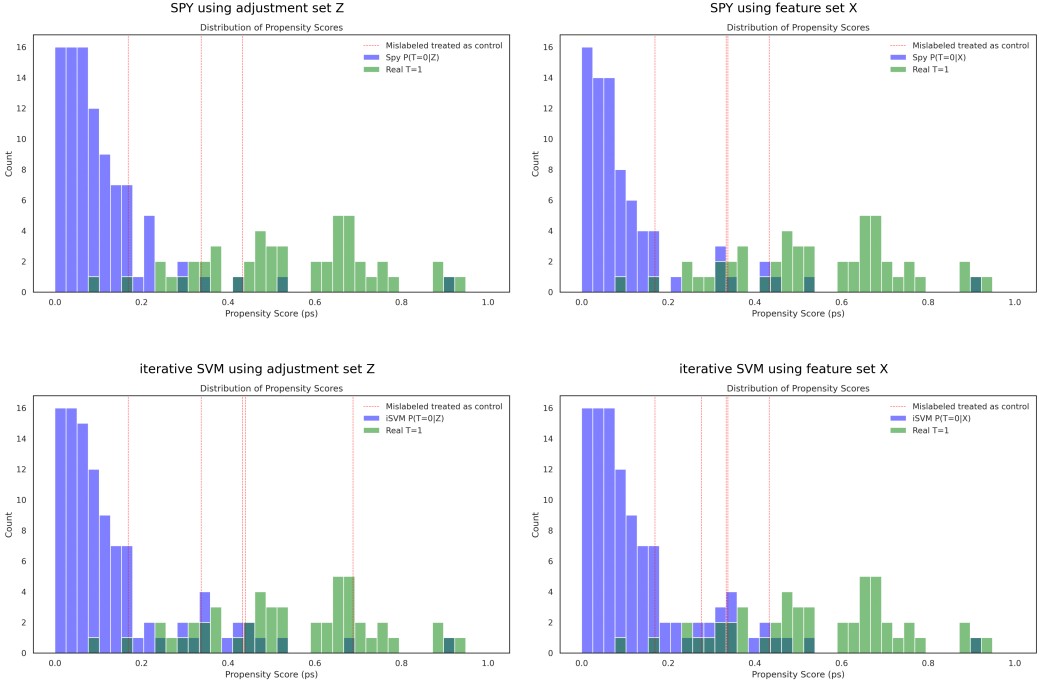

Figure 7: Propensity scores of 4 different combinations on experimental dataset regarding optimal sowing

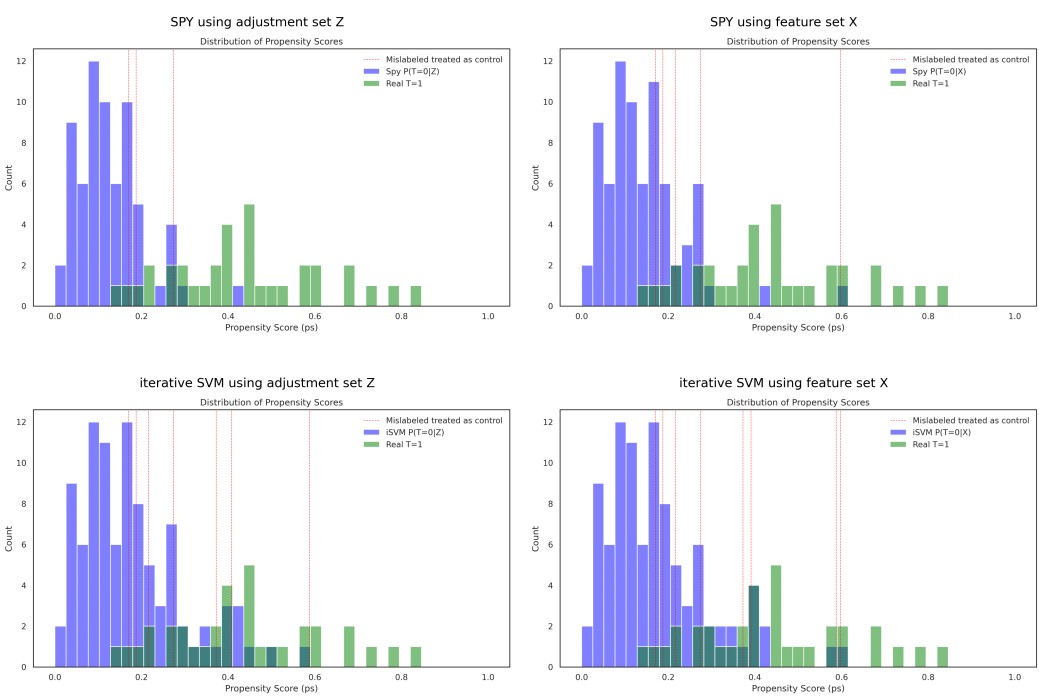

Figure 8: Propensity scores of 4 different combinations on experimental dataset regarding digestate fertilization application

# D  Interpretability and Model Comparison

To investigate and compare the internal decision mechanisms of two Support Vector Machine (SVM) classifiers, we examine the learned feature coefficients from each model. Both classifiers share a common architecture but differ in the feature sets used for training. We leverage the linear nature of the SVM (with a linear kernel) to extract and interpret feature importance directly from the model coefficients. These coefficients represent the influence of each feature on the decision boundary—positive values contribute toward predicting the positive class (e.g., sowing on a good day), and negative values contribute toward the negative class.

To visually communicate how feature importance shifts between the two models, we use a slope chart. Each line in the chart corresponds to a feature, connecting its normalized coefficient in the model trained on adjustment set $Z$ (left axis) to its corresponding value in the model trained of the superset of it, the feature set $X$ (right axis). Features that are newly introduced in the superset $X$ appear at zero on the left and are highlighted to emphasize their emergence in the decision function. Such comparative interpretability enhances our understanding of how additional information reshapes the model's decision boundary..

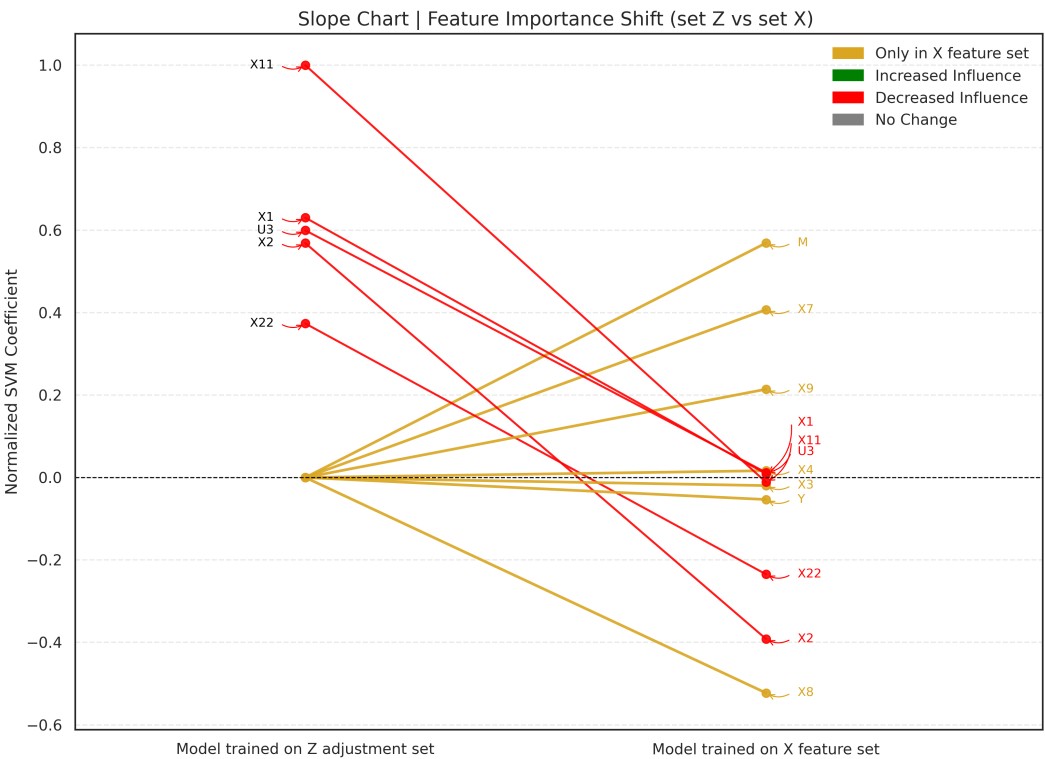

Figure 9: Interpretability and Model Comparison via SVM Coefficient Slope Chart for linear dataset

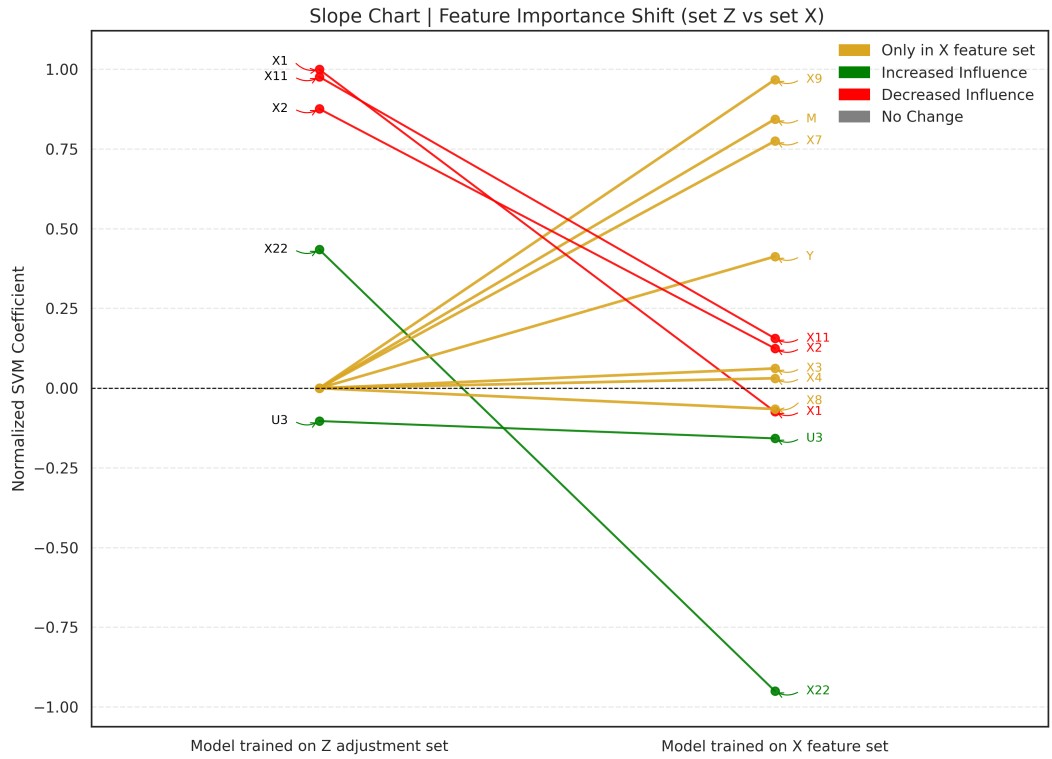

Figure 10: Interpretability and Model Comparison via SVM Coefficient Slope Chart for non-linear dataset

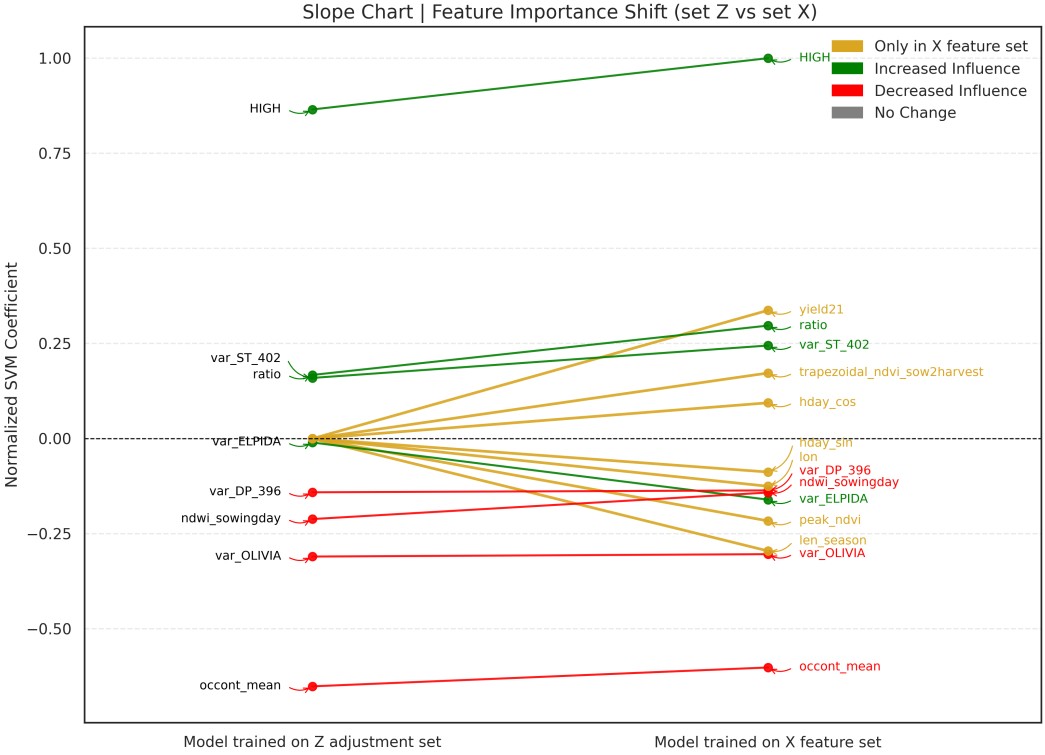

Figure 11: Interpretability and Model Comparison via SVM Coefficient Slope Chart for dataset regarding optimal sowing

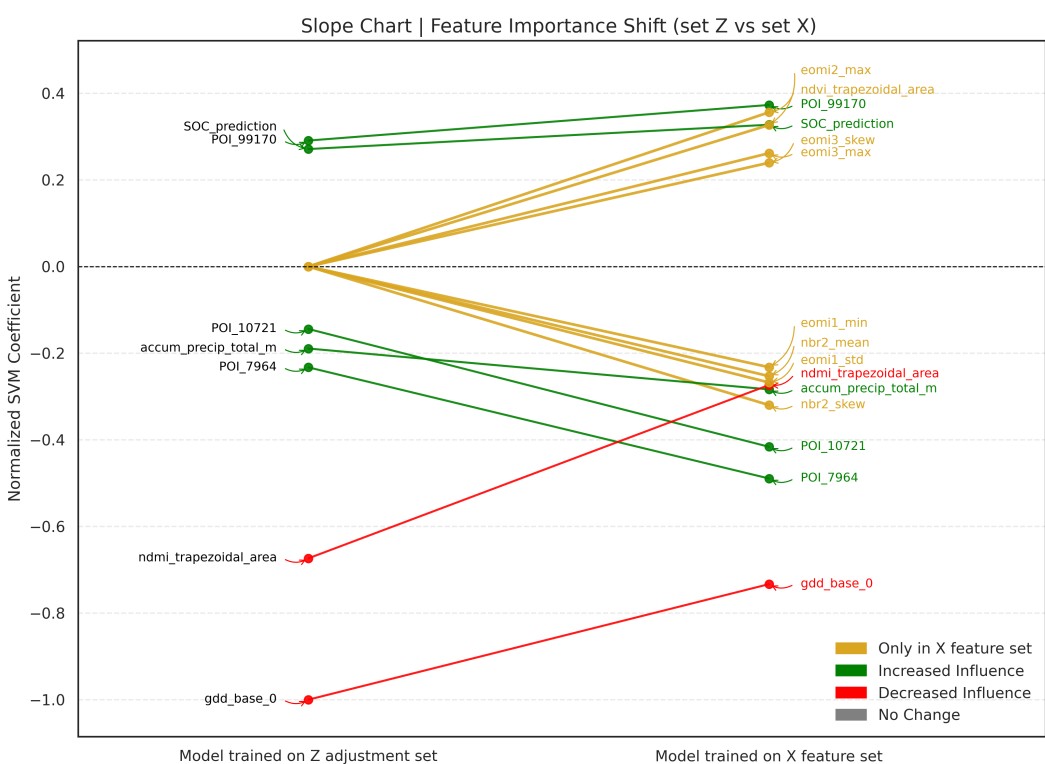

Figure 12: Interpretability and Model Comparison via SVM Coefficient Slope Chart for dataset regarding digestate fertilization application

