# OpenReview forum: "Positive-Unlabeled Learning for Control Group Construction in Observational Causal Inference"
_NeurIPS.cc/2025/Workshop/Reliable_ML — NeurIPS 2025 - Reliable ML Workshop_

### Official Review · Reviewer_bGKj · 2025-09-10
**Interesting idea and well written paper**

**Rating:** 8
**Confidence:** 3

**Review:**

**Summary**

The paper addresses the challenge of causal effect estimation when clearly labeled control units are absent, a common issue in observational studies. The authors propose leveraging Positive-Unlabeled (PU) learning to recover reliable control units from unlabeled populations, using only treated (positive) examples. They combine SPY and iterative SVM methods to identify controls and evaluate their framework on both simulated datasets (linear and nonlinear causal graphs) and real-world agricultural data (sowing and fertilization). The results show that PU learning can approximate control groups sufficiently well to yield reasonable ATE estimates, often close to those obtained using ground-truth controls.

**Strengths**

*Novelty & Relevance:* Applies PU learning in a causal inference context where no labeled controls exist—a relevant problem in fields like agriculture and Earth sciences where experiments are costly.

*Flexibility:* Unlike prior work (e.g., Kato et al. 2025), the method decouples control group construction from downstream effect estimation, allowing use with multiple causal estimators.

*Evaluation Design:* Includes both simulations (with known ground-truth ATEs) and real-world agricultural datasets, enhancing credibility.
*Clarity of Motivation:* The paper convincingly motivates the problem, emphasizing the practical importance of identifying controls in observational causal inference.

*Writing:* the paper is well written, with clear figures, tables and plots.

**Weaknesses / Limitations**

Stability on Small Datasets: Results in the fertilization dataset are unstable, with potential concerns about robustness.
Comparison Baselines: Limited benchmarking against alternative approaches for handling missing/contaminated controls.

**Suggestions for Authors**

Compare against recent alternative methods for missing control information, and consider running experiments on a larger real-world dataset.

**Overall Assessment:**
This is a relevant and creative application of PU learning to an underexplored causal inference problem. The work is promising and appropriate for a workshop setting but would benefit from deeper evaluation of robustness and comparisons.

---

### Official Review · Reviewer_7hSW · 2025-09-20
**The experiment results are interesting, but the writing/presentation can be improved. It is also hard to judge novelty.**

**Rating:** 6
**Confidence:** 3

**Review:**

Summary: This paper explores the use of PU learning as a preparatory step for causal estimation tasks (ATE estimation) in cases where one lacks access to labeled control group information. The work proposes using a classic method from PU learning, which is to train a classifier to distinguish between a truly treated (T = 1) and truly control (T = 0) unit. The data is process using the popular “hide and seek” approach where the positive-unlabeled dataset is constructed by transforming the original positive-negative dataset by hiding a percentage of positive/treated units in the negative/control units (SPY). The classifier used in this work is Naive Bayes and a second-stage iterative SVM is also considered, and experiments compare ATE estimation with and without this PU preprocessing step for several estimators.

Strengths:
- The experimental results indicate a promising direction of inquiry, where using PU learning as a preparatory step can help with ATE estimation in situations where one lacks control groups, showing that sometimes one can recover an ATE with this step that is close to the ATE estimated using confirmed treated and control groups.

Weaknesses:
- The presentation and writing could be improved, independent of the content.
- I think a more thorough or clear discussion of what this work newly contributes to the literature (over the prior/related work cited) is needed. It is unclear from reading the few sentences in Line 41, 42 what new ideas this work contributes.

Misc/suggestions:
- Table 1, 2 are a bit hard to parse and is written in very small font. Can the results be separated either into different tables or just some visualizations? Or can key results be bolded/highlighted?
- In general, rather than writing long blocks/paragraphs of text, summarizing key points into different paragraphs (with titles or other marker of separation) would make the paper easier to read
- Instead of using an entire page for Figures 3 and 4, perhaps it would be a better use of space to increase the font sizes in these figures and make the plots a bit smaller vertically. In general, I feel like the figures can be smaller (like Figure 2) but the tables should be a bit more readable.
- The equations (5)-(9) might look better if you align the terms or the definitions, as the current format looks a bit disorganized. You can even start a new paragraph/sentence on a new line which reads “We make the following assumptions:” or something like this.
- In Section 3.1, when describing the entire procedure including the classification step, perhaps a pseudo-code algorithm or informal algorithm but which concisely lists the steps in order, would be better for presentation

---

### Official Review · Reviewer_3VKX · 2025-09-20

**Rating:** 6
**Confidence:** 1

**Review:**

## Summary
The paper proposes positive-unlabeled (PU) learning to address the problem of the absence of clearly labeled control units in observational studies.

## Strengths and Weaknesses
### Strengths
The paper is easy to follow, and the writing is clean -  I hardly find any typos in this draft. The proposed method also sounds good to me.

### Weaknesses
Given this is a workshop submission, I hardly find any concerns with this draft. However, I have to admit that I am no expert in causality, so it is hard for me to judge this paper in depth, and I would leave the decision to other reviewers and the workshop organizers.

## Final comments
I think the draft is sound. However, since I am no expert in this field, I can only give this draft lukewarm support.